# Determining relative population-specific acceleration intensity thresholds in soccer using game locomotion data: Validation of a new method using data from male youth elite players

Pascal Andrey[1]*, Karin Fischer-Sonderegger[1], Wolfgang Taube[2], Markus Tschopp[1]

**1** Department of Elite Sport, Swiss Federal Institute of Sport Magglingen SFISM, Magglingen, Bern, Switzerland, **2** Department of Neurosciences and Movement Science, University of Fribourg, Fribourg, Switzerland

* pascal.andrey@baspo.admin.ch

## Abstract

In soccer, relative population-specific acceleration intensity thresholds are required to create meaningful activity profiles. These thresholds can be derived from the *maximal acceleration-initial running speed ($a_{max}$-$v_{init}$) regression line*, whose determination has so far required time-consuming testing. The aims of this study were to introduce a new method for determining population-specific $a_{max}$-$v_{init}$ regression lines in soccer using game locomotion data and to assess its validity as a function of the amount of data used. The method accounts for both the amount of data used and the distribution of high-intensity accelerations across the velocity measurement range when identifying maximal accelerations in game locomotion data. This is intended to minimize the risk of selecting submaximal accelerations or predominantly maximal accelerations with a positive random measurement error. Game locomotion data were collected from 55 male youth elite soccer players using a GPS-based tracking system. Multiple population-specific $a_{max}$-$v_{init}$ regression lines were determined using locomotion data from one to five games per athlete. Furthermore, each athlete completed an acceleration test to determine his *test-based $a_{max}$-$v_{init}$ regression line*. The mean biases for the regression coefficients (i.e., $a_{max}$-intercept and slope) were estimated and assessed using standardization and Bayesian analysis. Regression lines based on locomotion data from two or three combined games showed trivial biases for both coefficients. However, due to the large uncertainty in the estimates, the chance of equivalence was only assessed as *possibly equivalent*. The proposed *game-based* method represents a viable and easy-to-implement alternative to the test-based method for determining population-specific $a_{max}$-$v_{init}$ regression lines in soccer. This simplifies the process of determining relative population-specific acceleration intensity thresholds, which are required for creating meaningful activity profiles.

**Data availability statement:** The data that support the findings of this study and the code used for the statistical analysis are openly available in figshare at https://doi.org/10.6084/m9.figshare.25927285.v1.

**Funding:** The author(s) received no specific funding for this work.

**Competing interests:** The authors have declared that no competing interests exist.

## Introduction

In soccer, tracking technologies are widely used to assess players' activity during training sessions and matches [1,2]. However, describing the complex activity of soccer in a meaningful way using locomotion data is a major challenge [3–5].

In terms of locomotion, soccer is characterized by frequent changes in running speed [6], necessitating the use of acceleration-based metrics to create meaningful activity profiles [7]. A commonly applied metric in both scientific research and practical training settings is the number of accelerations exceeding certain intensity thresholds, with absolute generic thresholds being predominantly employed. For instance, thresholds of 2 m·s$^{-2}$ or 3 m·s$^{-2}$ are frequently used to classify an acceleration action as highly intense [8–10]. However, reliance on such absolute generic thresholds presents several limitations. First, maximal reachable acceleration decreases as running speed increases [5]. Consequently, the use of an absolute intensity threshold leads to an overestimation of the intensity of acceleration actions initiated from a stationary position or low running speeds, while underestimating the intensity of those initiated from higher speeds [3,5]. Second, soccer players from different populations (e.g., from different performance levels, age categories, or genders) exhibit differences in their neuromuscular performance capacity and hence in their maximal acceleration capacity [11–16]. As a result, a generic intensity threshold may overestimate the intensity of acceleration actions in populations with higher maximal acceleration capacity (i.e., elite athletes) while underestimating it in populations with lower maximal acceleration capacity (i.e., amateur athletes) [4].

Therefore, absolute generic intensity thresholds do not allow for a valid intensity assessment of acceleration actions in soccer [3–5]. An alternative approach involves the use of *relative population-specific intensity thresholds* [5]. These thresholds assess the intensity of an acceleration action in relation to the maximal acceleration that can, on average, be reached within a given population from a certain initial running speed. This approach enables a more valid assessment of acceleration intensity in soccer and thus to the creation of meaningful activity profiles [17,18].

Relative population-specific acceleration intensity thresholds can be derived from a population-specific *maximal acceleration-initial running speed ($a_{max}$-$v_{init}$) regression line* [5]. The current reference method for determining this regression line involves a performance test in which athletes complete four maximal accelerations from different initial running speeds [5]. However, a recent study by Silva et al. [19] introduced an alternative approach, determining a population-specific $a_{max}$-$v_{init}$ regression line in soccer players using training locomotion data collected via global positioning system (GPS) tracking rather than performance test data. The use of automatically recorded locomotion data from training sessions and/or matches offers the significant advantage of not imposing additional expenditures of time or physical load on the athletes. Furthermore, this approach allows for the utilization of large existing datasets from competitive matches, such as those collected during tournaments or championships, to derive truly population-specific $a_{max}$-$v_{init}$ regression lines. However, before replacing an established reference method with a new approach, rigorous validation is

required [20]. In their methodological study, Silva et al. [19] did not examine the validity of their $a_{max}$-$v_{init}$ regression derived from training locomotion data.

In our view, the major methodological challenge in determining the $a_{max}$-$v_{init}$ regression line of soccer players using training or game locomotion data is to identify the maximal accelerations from the totality of accelerations. Silva et al. [19] propose a straightforward approach in which all acceleration actions are plotted in an $a_{max}$-$v_{init}$ diagram, and the highest accelerations are selected across the $v_{init}$ measurement range. The selection process is characterized by the fact that a predefined number of accelerations is selected and that the selection is made evenly distributed across the $v_{init}$ measurement range. However, this approach presents two major limitations. First, if a dataset contains fewer maximal accelerations than the predefined number to be selected, submaximal accelerations will inevitably be included. While Silva et al. [19] suggest addressing this issue by utilizing a sufficiently large amount of locomotion data, this approach warrants careful consideration due to the substantial magnitude of the random measurement error associated with GPS-based tracking systems when measuring maximal acceleration [21,22]. Measurement inaccuracy is an inherent an unavoidable aspect of every measurement taken by human-created measuring instruments [23,24]. Consequently, when selecting only the highest recorded value from a large number of repeated measurements of a given variable (in this case, multiple maximal accelerations), it is highly probable that the selected value includes a positive random measurement error [23,24]. Therefore, we assume that in the approach of Silva et al. [19], using a large amount of locomotion data may result in the inclusion of predominantly maximal accelerations with a positive random measurement error, ultimately leading to an overestimation of an individual's or a population's true maximal acceleration capacity.

The second limitation arises from selecting accelerations evenly distributed across the $v_{init}$ measurement range. In soccer, however, maximal accelerations are not evenly distributed across this range but are instead right-skewed, meaning that a greater number of maximal accelerations occur at lower $v_{init}$ values compared to higher $v_{init}$ values [17,18,25]. As a result, in the low $v_{init}$ measurement range, acceleration selection is performed from a larger pool of maximal accelerations than in the high $v_{init}$ measurement range. This discrepancy is important to consider, as a greater number of repeated measurements of a given variable (in this case, maximal accelerations) increases the likelihood for measurements with a large random measurement error [23,24]. Therefore, we hypothesize that selecting accelerations evenly across the $v_{init}$ measurement range results in the inclusion of predominantly maximal accelerations with a positive random measurement error in the low $v_{init}$ range. Conversely, in the high $v_{init}$ range, the selection is more likely to include either maximal accelerations with a lower positive or negative random measurement error or even submaximal accelerations. This potential bias could lead to an overestimation of the slope of an athlete's or population's $a_{max}$-$v_{init}$ regression line (i.e., too steep a slope).

While we acknowledge that the approach proposed by Silva et al. [19]—using training or game locomotion data to determine $a_{max}$-$v_{init}$ regression lines—is both innovative and valuable, we believe that the methodology can be further refined. Therefore, the aims of this study were (1) to introduce a new method for determining the $a_{max}$-$v_{init}$ regression line of soccer players using game locomotion data and (2) to investigate the validity of population-specific $a_{max}$-$v_{init}$ regression lines derived using this new method across varying amounts of game locomotion data.

## Materials and methods

### Participants

A total of 55 male youth elite soccer players from six Swiss under-18 and under-21 elite teams participated in the study (age 18.2 ± 2.3 y, height 178.6 ± 5.6 cm; weight 73.6 ± 7.2 kg). Goalkeepers were excluded. The study was conducted in accordance with the Declaration of Helsinki and approved by the Ethics Committee of Bern (Project ID: 2019–01586, 19 November 2019). The teams and their players were recruited between May 1, 2021 and July 31, 2021. Players received verbal and written information about the study design before giving written informed consent. The ethics committee waived the need for parental or guardian consent for minor players.

## Design

For validity testing, a cross-sectional study design was used. During the 2021/2022 soccer season, locomotion data from all official championship matches were collected. Midway through the second half of the season, the participants took an acceleration test to determine their $a_{max}$-$v_{init}$ regression line [5]. One week before the actual test, a familiarization test was performed. The *test-based regression line* served as the criterion measure. The *game-based regression line* was determined based on the locomotion data of the game closest to the test. In determining a game-based regression line based on data from multiple games, the games closest to the test were always combined. For each athlete, only games in which he had participated for at least 80 minutes were used.

## Measurements

All games and the acceleration test were recorded with a 10 Hz GPS-based tracking system (Advanced Sport Instruments, FieldWiz V2, Lausanne, Switzerland). To prevent measurement errors due to possible inter-unit variation, each athlete always wore the same sensor for all measurements. The GPS units derived instantaneous velocity via the Doppler shift method. Unfortunately, the number of connected satellites and the horizontal dilution of precision during the measurements are not provided by the manufacturer for FieldWiz V2 devices. The validity of 10 Hz GPS devices has been examined in several studies. For the instantaneous velocity, typical errors of the estimate (TEE) between 0.12 and 0.32 m·s$^{-1}$ [21,26,27] or 3.1 and 11.3% [28] were reported. For the maximal acceleration in sprints, the TEE values were between 0.27 and 0.33 m·s$^{-2}$ [21].

The acceleration test was performed according to the protocol of Sonderegger et al. [5], consisting of four maximal accelerations from different initial running speeds. Reliability testing of the maximal acceleration from the different initial running speeds revealed statistically non-significant ($p > 0.05$), trivial to small (standardized effect size <0.4) differences in means between test and retest [5]. To minimize the random measurement error in the test-based $a_{max}$-$v_{init}$ regression line, only regression lines with an $R^2 \geq 0.90$ were included in the analysis (mean ± SD $R^2$ test-based regression lines: 0.96 ± 0.03). All tests were performed at the training site of the respective clubs. Before the test, the athletes performed a standardized warm-up consisting of mobility and stability exercises and dynamic warm-up exercises and finished with four maximal accelerations from different initial running speeds. On the test day, athletes were in a recovered state (i.e., match day +3 or +4 and no intensive training session the day before).

## Data analysis

**Data processing and event detection.** The Doppler shift velocity signal and the time stamp of all measurements were exported from FieldWiz online software. All analyses were then performed based on these data using a custom MATLAB script (Version 9.8.0 (R2020a), MathWorks Inc., Natick, USA). According to the manufacturer, the exported velocity signal had already been smoothed with a one-second moving average filter. Therefore, no further filtering techniques were applied. The acceleration signal was calculated as the first derivative of the velocity signal. The two signals were then used to detect acceleration actions, applying the same procedure as Fischer-Sonderegger et al. [17]. Each action was described by its initial running speed ($v_{init}$) and the maximal acceleration reached ($a_{max}$).

**Event selection and model fitting.** To determine the $a_{max}$-$v_{init}$ regression line of an athlete based on the totality of detected acceleration actions in one or multiple games, maximal acceleration actions from different initial running speeds must be identified and subsequently described by a linear model [5]. For this purpose, when using locomotion data from one game, the following procedure was applied (see Fig 1 for a graphical illustration).

(1) All detected acceleration actions were plotted in an $a_{max}$-$v_{init}$ diagram.

(2) The $v_{init}$-axis of the diagram was divided into intervals of a length of 4.3 km·h$^{-1}$.

(3) Within each interval, the action with the highest $a_{max}$ was selected.

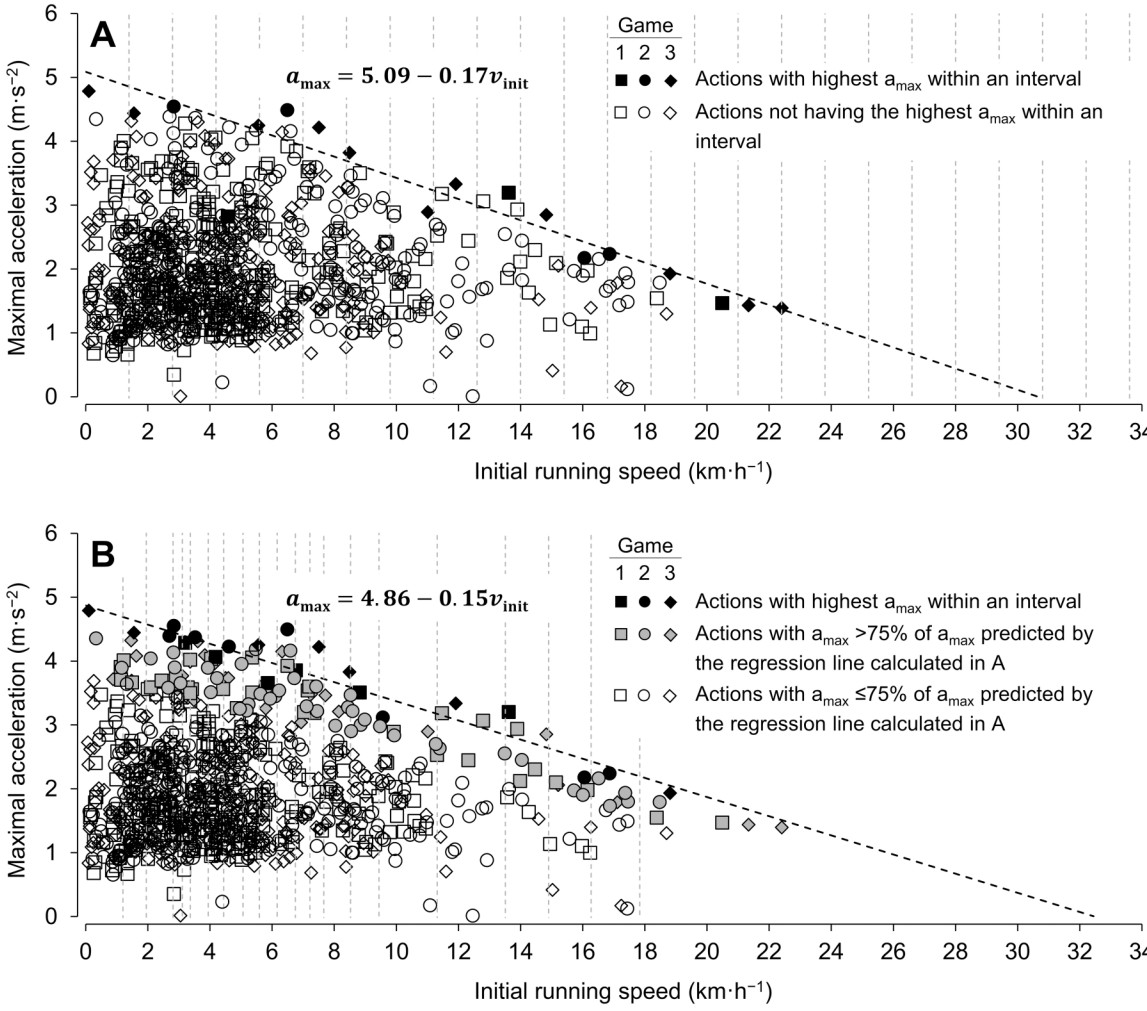

**Fig 1. Procedure for determining a maximal acceleration-initial running speed ($a_{max}$-$v_{init}$) regression line using game locomotion data.** The figure shows the procedure for the case of three combined games. Different symbols represent acceleration actions from different games for a representative athlete. (A) (1) The $v_{init}$-axis is divided into intervals with a length of 1.4 km·h⁻¹ (gray dashed vertical lines), (2) within each interval the action with the highest $a_{max}$ is selected (black filled symbols), and (3) the selected actions are described by a linear model (black dashed line). (B) (1) High-intensity acceleration actions (i.e., actions with an $a_{max}$ > 75% of the $a_{max}$ predicted by the regression line calculated in (A)) are selected (gray filled symbols), (2) 20 quantiles are calculated for the $v_{init}$-values of the high-intensity acceleration actions (gray dashed vertical lines), (3) within each interval between successive quantiles, the action with the highest $a_{max}$ is selected (black filled symbols), and (4) the actions selected in this way are described by a linear model, resulting in the $a_{max}$-$v_{init}$ regression line of the athlete (black dashed line).

(4) The selected actions were described by a linear model using a robust regression technique [29] (Fig 1A).

(5) The estimated regression line was used to select all high-intensity actions (i.e., those with an $a_{max}$ > 75% of the $a_{max}$ predicted by the regression line).

(6) Six quantiles were calculated for the $v_{init}$-values of the high-intensity actions.

(7) Within each interval between successive quantiles, the action with the highest $a_{max}$ was selected.

(8) The selected actions were again described by a linear model using a robust regression technique [29]. The regression line of this analysis shows the $a_{max}$-$v_{init}$ regression line of an athlete (Fig 1B).

When the $a_{max}$-$v_{init}$ regression line of an athlete was determined based on locomotion data from multiple games, the interval length and the number of quantiles were adjusted accordingly (Table 1). Specifically, the larger the amount of locomotion data used (e.g., the more data from different games combined), the shorter the interval length was set and the more quantiles were calculated. This results in more acceleration actions being selected for the regression analyses with an increasing amount of locomotion data used. The use of quantile-based intervals ensures that the acceleration actions selected for the final regression analysis are each selected from an equally large set of high-intensity actions.

The athletes' test-based $a_{max}$-$v_{init}$ regression lines were determined according to the reference method of Sonderegger et al. [5] by a linear least squares regression analysis of the four acceleration actions recorded in the test. From both regression lines (game- and test-based), the coefficients of the regression equation ($a_{max}$-intercept and slope) were then used for the statistical analysis.

## Statistical analysis

The measures of centrality and dispersion are the mean ± *SD*. Statistical modeling was performed using SAS Studio (Version 3.81; SAS Institute Inc., Carry, USA). Due to non-uniform effects and errors of groups formed based on playing positions, the mixed linear modeling procedure (PROC MIXED) was used [30]. Mixed linear models allow for estimating different effects and errors in different groups, which can subsequently be combined into a single effect estimate. With this approach, bias and imprecision in an effect estimate due to non-uniform effects or errors within subgroups can be avoided [30,31]. The reason for the observed non-uniform effects and errors could be the different activity profiles of the different playing positions in games (i.e., different numbers of maximal accelerations and different distributions of maximal accelerations across the $v_{init}$ measurement range) [17,18,25,32].

In the statistical model, the coefficients of the $a_{max}$-$v_{init}$ regression equations were the outcome variables, and a separate analysis was performed for each. The fixed effect was playing position × measurement condition (to estimate the mean of the outcome variable for the different measurement conditions [i.e., test and game] within groups formed by playing positions [center-backs, central midfielders, forwards, and outside players [i.e., full-backs and wide midfielders combined]]). The random effect was the intercept (to account for differences in the outcome variable between players). Player identity was the subject variable (to model the nesting of measurements within players), and playing position was the group variable (to model the nesting of players within positions). The estimated position-specific means of the outcome variable were then combined to obtain a single mean for the test condition, the game condition, and the bias (i.e., game − test) (using an LSMESTIMATE statement). In this combination, the outside players' mean was weighted double that of the other position-specific means. This results in effects that are similar to those of a fully pooled analysis (i.e., only the measurement condition as a fixed effect in the model) if the sample contained the same number of center-backs, central midfielders, and forwards and a double number of outside players. Thus, the position groups were weighted according to their proportions in the population. The mean bias in raw, percentage, and standardized units was the measure of validity. Standardization was performed using the between-subject *SD* of the test condition (i.e., the criterion measure).

**Table 1. Interval length and number of quantiles as a function of the number of games combined.**

| Number of games | Interval length (km·h⁻¹) | Number of quantiles |
|---|---|---|
| 1 | 4.3 | 6 |
| 2 | 2.1 | 13 |
| 3 | 1.4 | 20 |
| 4 | 1.1 | 27 |
| 5 | 0.9 | 34 |

Plots of residuals versus predicted values showed no evidence of nonuniformity of error and no outliers (defined as observations with a studentized residual of >3.5). Uncertainty in the estimates of effects is presented as 90% compatibility intervals (CI). Analogous to the effect estimates, CIs were obtained by combining position-group-specific intervals [30,31]. Decisions about magnitudes of effects accounting for the uncertainty were based on a reference Bayesian analysis with a minimally informative prior [33,34], which provided estimates of chances that the true magnitude was a substantial negative value, a trivial value (i.e., game measure equivalent to the test measure), or a substantial positive value. For these calculations, a previously published spreadsheet was used [35].

All effects are reported with a qualitative descriptor for the chance of the effect to be trivial using the following scale: ≤0.005, *most unlikely*; >0.005–0.05, *very unlikely*; >0.05–0.25 *unlikely*; >0.25–0.75, *possibly*; >0.75–0.95, *likely*; >0.95–0.995, *very likely*; and >0.995, *most likely* [31,36]. Magnitudes of standardized effects were assessed as ≤0.2, *trivial*; >0.2–0.6, *small*; >0.6–1.2, *moderate*; >1.2–2.0, *large*; >2.0–4.0, *very large*; and >4.0, *extremely large* [31].

## Results

Table 2 shows the final sample sizes, descriptive statistics, and mean bias estimates in raw and percentage units. Fig 2 shows the standardized mean bias estimates, their magnitude, and the decision on equivalence. Compared to a single game, the combination of locomotion data from two games led to a reduction in the mean bias estimates of both regression coefficients ($a_{max}$-intercept and slope). However, with the addition of further games, the mean bias for the $a_{max}$-intercept did not change meaningfully. In contrast, for the slope, the addition of a fourth and fifth game led to a new gradual increase in the mean bias. Thus, with trivial mean biases for both coefficients, the regression lines based on two and three combined games were equivalent to the test-based regression line. However, due to the large uncertainty in the estimates, the chance that the true effect of the coefficients of these two regression lines is trivial, or that the coefficients are indeed equivalent to the coefficients of a test-based line, was only assessed as *possibly trivial* or *equivalent*.

## Discussion

Relative acceleration intensity thresholds are essential for creating meaningful activity profiles in soccer, such thresholds can be derived from an $a_{max}$-$v_{init}$ regression line. In this study, we introduced a new method for determining the $a_{max}$-$v_{init}$ regression line of soccer players using game locomotion data. This method accounts for both the amount of locomotion data used and the individual distribution of high-intensity accelerations across the $v_{init}$ measurement range when identifying maximal accelerations. By incorporating these factors, the proposed approach aims to minimize the risk of selecting submaximal accelerations or predominantly maximal accelerations with a positive random measurement error, thereby enhancing the validity of an athlete's $a_{max}$-$v_{init}$ regression line. The advantage of a game-based method over the test-based approach is its ease of implementation, as it eliminates the need for time-consuming performance testing.

Table 2. Descriptive statistics and mean bias estimates in raw and percent units.

| Games[a] | n | Mean game-based coefficients (*SD*) | | Mean test-based coefficients (*SD*) | | Mean bias raw units [90%CI] | | Mean bias percent units [90%CI] | |
|---|---|---|---|---|---|---|---|---|---|
| | | $a_{max}$-intercept | Slope | $a_{max}$-intercept | Slope | $a_{max}$-intercept | Slope | $a_{max}$-intercept | Slope |
| 1 | 41 | 4.65 (0.62) | −0.128 (0.081) | 4.85 (0.32) | −0.158 (0.025) | −0.20 [−0.36, −0.03] | 0.030 [0.007, 0.052] | −4.06 [−7.42, −0.70] | 18.96 [4.74, 33.19] |
| 2 | 32 | 4.85 (0.33) | −0.152 (0.033) | 4.85 (0.34) | −0.156 (0.028) | 0.00 [−0.12, 0.11] | 0.004 [−0.007, 0.016] | −0.04 [−2.44, 2.35] | 2.86 [−4.40, 10.12] |
| 3 | 29 | 4.86 (0.28) | −0.151 (0.025) | 4.83 (0.36) | −0.155 (0.028) | 0.03 [−0.08, 0.14] | 0.004 [−0.007, 0.014] | 0.61 [−1.66, 2.89] | 2.28 [−4.59, 8.80] |
| 4 | 28 | 4.83 (0.22) | −0.145 (0.015) | 4.82 (0.35) | −0.153 (0.027) | 0.02 [−0.09, 0.12] | 0.008 [−0.001, 0.017] | 0.33 [−1.92, 2.57] | 5.12 [−0.96, 11.20] |
| 5 | 23 | 4.84 (0.21) | −0.143 (0.017) | 4.85 (0.36) | −0.155 (0.029) | −0.01 [−0.14, 0.13] | 0.012 [0.000, 0.024] | −0.18 [−2.94, 2.58] | 7.74 [0.06, 15.41] |

[a]Number of games combined for determining the game-based regression line.

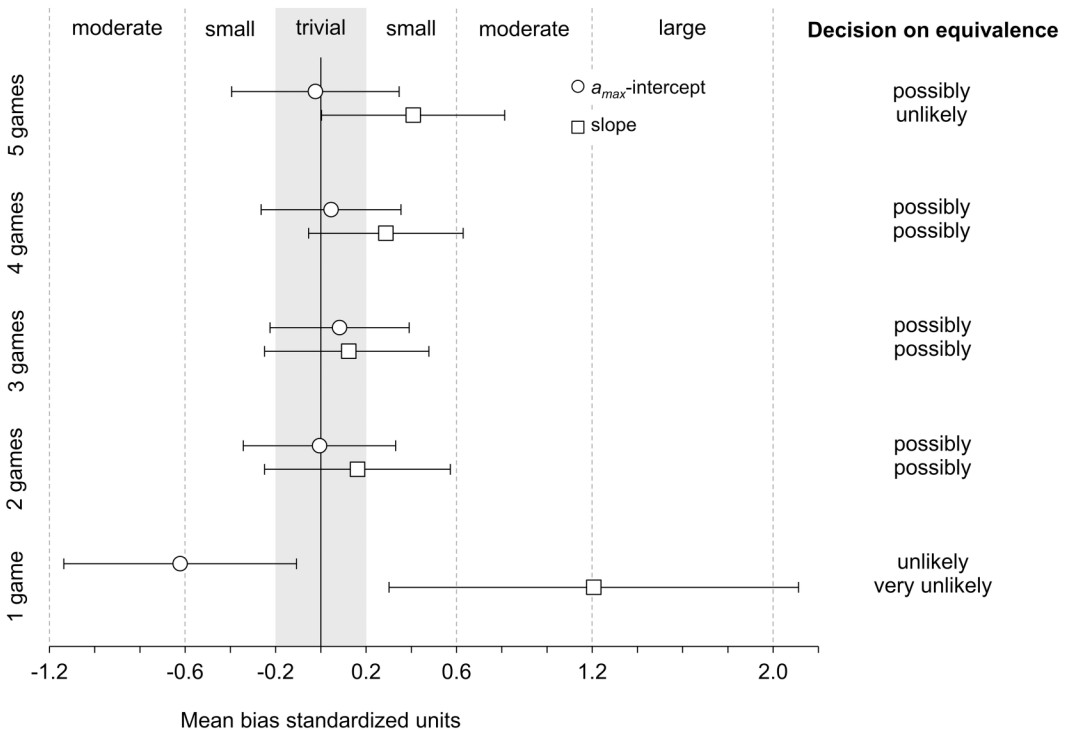

**Fig 2. Standardized mean bias estimates and the respective decision on equivalence.** The gray shaded area shows the trivial effect or equivalence range. Error bars show 90% compatibility intervals.

Using this new method with varying amounts of game locomotion data from male youth elite soccer players, we determined population-specific regression lines and compared them with a test-based regression line. Game-based regression lines based on locomotion data from two or three games differed only trivially from the test-based line. However, due to the large uncertainty in the estimates, the chance that regression lines based on two or three combined games are indeed equivalent to a test-based line was only assessed as *possibly equivalent*. Despite this limitation, in scenarios where determining a test-based, population-specific $a_{max}$-$v_{init}$ regression line is not feasible, the proposed game-based method represents a viable alternative.

The method presented in this study for determining the $a_{max}$-$v_{init}$ regression line of soccer players differs in several aspects from a previously published approach [19]. A first distinction is that our method accounts for the amount of locomotion data used (i.e., the number of combined games) when selecting accelerations for the regression analyses. The increase in the number of accelerations to be selected with an increasing number of combined games is designed to reflect the likely increase in the number of maximal accelerations as the dataset expands. This methodological consideration is important because if the number of accelerations to be selected (i.e., the assumed number of maximal accelerations) does not correspond to the (unknown) true number of maximal accelerations in the dataset, the resulting regression line is likely to be biased. Specifically, if the number of accelerations to be selected is larger than the actual number of maximal accelerations, submaximal accelerations will inevitably be selected. This would result in an underestimation of an athlete's true regression line (i.e., a too-low $a_{max}$-intercept). Conversely, if the number of maximal accelerations in a dataset starts to exceed the number of accelerations to be selected, the applied acceleration selection procedure (e.g., selecting only the highest accelerations across the $v_{init}$ measurement range) increasingly selects only maximal accelerations with a positive random measurement error (see Rabinovich [23] and Hopkins [24] regarding the concept of

random measurement error). This would result in an overestimation of an athlete's true regression line (i.e., a too-high $a_{max}$-intercept). Furthermore, an $a_{max}$-$v_{init}$ regression line based primarily on maximal accelerations with a positive random measurement error is also expected to exhibit a steeper slope than the true regression line. This assumption is due to the larger magnitude of the random measurement error in the low $v_{init}$ region compared to the high $v_{init}$ region [22].

A second difference is that our method accounts for the individual frequency distribution of high-intensity accelerations across the $v_{init}$ measurement range when selecting maximal accelerations. In practical terms, a greater number of accelerations are selected in regions of the $v_{init}$ measurement range where an athlete exhibited a higher frequency of high-intensity accelerations. The rationale for this methodological approach is that we assume that the frequency distribution of maximal accelerations closely approximates, or ideally matches, the frequency distribution of high-intensity accelerations. An incorrect assumption regarding the frequency distribution of maximal accelerations across the $v_{init}$ measurement range inevitably leads to two problems. First, in regions where the assumed frequency of maximal accelerations is too low, primarily maximal accelerations with a positive random measurement error are selected [23,24]. Second, in regions where the assumed frequency of maximal accelerations is too high, either maximal accelerations with a lower positive or negative random measurement error are selected, or it comes to the selection of submaximal accelerations. These mechanisms would cause a bias in the slope of the regression line.

In summary, the method presented in this study accounts for both the amount of locomotion data used and the individual distribution of high-intensity accelerations across the $v_{init}$ measurement range when identifying maximal accelerations. This approach is designed to minimize the risk of selecting submaximal accelerations or predominantly maximal accelerations with a positive random measurement error. In other words, the method aims to identify all maximal accelerations within a data set, both those with a positive and those with a negative random measurement error. Given the substantial magnitude of the random measurement error associated with GPS-based tracking systems when measuring maximal acceleration [21,22], we consider this refinement essential for determining a valid $a_{max}$-$v_{init}$ regression line.

This study is the first to assess the validity of a population-specific $a_{max}$-$v_{init}$ regression line determined using game or training locomotion data. In our study, the regression lines based on locomotion data from two or three games were the most similar to the test-based regression line. For these regression lines, both the mean bias estimate for the $a_{max}$-intercept and the slope were of trivial magnitude. However, due to the large uncertainty in the estimates, the chance for equivalence was only assessed as *possibly equivalent*. The large uncertainty in the estimates also prevents a definitive conclusion on whether two and three combined games really is an optimal amount of data, or whether the observed differences between the mean bias estimates across regression lines are attributable to sampling variation. In principle, we interpret the mostly trivial or small mean bias estimates observed in this study to mean that the assumptions made in the method regarding the number and distribution of maximal accelerations were on average appropriate for the underlying data. However, the large uncertainty in the estimates indicates that these assumptions were not suitable for every individual athlete. As a result, meaningful differences between the game-based and test-based regression lines were likely present at the individual level.

## Practical application

Relative population-specific acceleration intensity thresholds account for both the initial running speed and the performance capacity of a given population when assessing the intensity of an acceleration action. By doing so, they overcome the limitations of absolute generic acceleration intensity thresholds and enable a more valid intensity assessment of acceleration actions in soccer, thereby facilitating the creation of meaningful activity profiles [3,17,18]. This, in turn, allows for a more accurate estimation of training and match load, as well as a deeper understanding of the performance demands in soccer [37]. The game-based method presented in this study can be considered a viable and easy-to-implement alternative to the test-based method [5] for determining $a_{max}$-$v_{init}$ regression lines in additional populations of soccer players (e.g., women, different age categories, or varying performance levels). This requires locomotion data from only two or three

games per athlete. From these regression lines, relative population-specific acceleration intensity thresholds can then be derived. For instance, a regression line corresponding to 75% of the $a_{max}$-$v_{init}$ regression line can be used as a threshold for high-intensity actions [5]. Since $a_{max}$-$v_{init}$ regression lines of other populations are currently lacking, the possibility of determining them with relatively little effort using game locomotion data is of significant practical value. Compared to the test-based approach, the game-based method facilitates the determination of regression lines based on data from a truly representative sample of a population (e.g., incorporating data from different clubs or leagues). Furthermore, the game-based approach is particularly advantageous for populations where conducting an acceleration test is impractical due to the additional time and physical load involved, such as elite soccer players.

## Conclusion

In this article, we present a new method for determining the $a_{max}$-$v_{init}$ regression line of soccer players using game locomotion data and discuss the associated challenges. This method accounts for both the amount of locomotion data used and the distribution of high-intensity accelerations across the $v_{init}$ measurement range when identifying maximal accelerations. By doing so, it minimizes the risk of selecting submaximal accelerations or predominantly maximal accelerations with a positive random measurement error. Our results indicate that this method is a viable alternative to the test-based approach for determining population-specific $a_{max}$-$v_{init}$ regression lines in soccer. Future studies can apply this method to determine $a_{max}$-$v_{init}$ regression lines for other populations with relatively little effort. Thus, the findings of this study significantly simplify the process of defining relative population-specific acceleration intensity thresholds in soccer. Such thresholds are essential for a valid intensity assessment of acceleration actions and, ultimately, for creating meaningful activity profiles.

## Acknowledgments

Lionel Castella and Nicco Vögeli both contributed to writing parts of the MATLAB script used to determine the game-based $a_{max}$-$v_{init}$ regression lines.

## Author contributions

**Conceptualization:** Pascal Andrey, Karin Fischer-Sonderegger, Wolfgang Taube, Markus Tschopp.

**Data curation:** Pascal Andrey.

**Formal analysis:** Pascal Andrey.

**Investigation:** Pascal Andrey.

**Methodology:** Pascal Andrey.

**Project administration:** Pascal Andrey.

**Resources:** Pascal Andrey, Markus Tschopp.

**Supervision:** Wolfgang Taube.

**Validation:** Pascal Andrey.

**Visualization:** Pascal Andrey.

**Writing – original draft:** Pascal Andrey.

**Writing – review & editing:** Pascal Andrey, Karin Fischer-Sonderegger, Wolfgang Taube, Markus Tschopp.

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
