## [Decision Letter · Decision Letter 0]

30 Jan 2025

PONE-D-24-58536Determining relative population-specific acceleration intensity thresholds in soccer using game locomotion data: Validation of a new methodPLOS ONE

Dear Dr. Andrey,

**We have received the comments from the two Reviewers, and after careful consideration, we feel that it has merit but does not fully meet PLOS ONE’s publication criteria as it currently stands. Given their suggestions, the paper's quality is overall good, but it requires some minor adjustments.**

Therefore, we invite you to submit a revised version of the manuscript that addresses the points raised during the review process.

We look forward to receiving your revised manuscript.

Kind regards,

Stefano Amatori, Ph.D.

Academic Editor

PLOS ONE

**Journal Requirements:**

Reviewers' comments:

Reviewer's Responses to Questions

**Comments to the Author**

1. Is the manuscript technically sound, and do the data support the conclusions?

Reviewer #1: Yes

Reviewer #2: Yes

2. Has the statistical analysis been performed appropriately and rigorously? 

Reviewer #1: Yes

Reviewer #2: Yes

3. Have the authors made all data underlying the findings in their manuscript fully available?

Reviewer #1: Yes

Reviewer #2: Yes

4. Is the manuscript presented in an intelligible fashion and written in standard English?

Reviewer #1: Yes

Reviewer #2: Yes

5. Review Comments to the Author

**Reviewer #1: ** The article aimed to present and validate a new method for determining population-specific acceleration intensity thresholds in soccer, using game locomotion data. The paper’s quality is overall good: the abstract is clear enough, the design has been well explained and the results are presented in a good form.

I have detailed below some additional comments that could help to further strengthen the manuscript quality.

Title:

The investigated population is composed by male youth elite soccer players. This should be indicated in title.

Introduction:

The introduction presents some sentences that are too short, fragmenting linearity of text. In addition, many phrases has no references, that should be added. In particular the following points require attention:

Line 42: Reference required

Lines 43 – 63: Sentences too short fragmenting linearity of text and too many phrases with no reference

Lines 67 – 70: Sentences too short fragmenting linearity of text and too many phrases with no reference

Line 97: Reference required

Lines 102 – 107: References required

Materials and methods:

Participants:

A table including descriptive statistics of anthropometric measures could be helpful to improve section clarity.

Measurements:

It should be clarified if warm up protocols before tests have been standardized.

Data analysis:

Event selection and model fitting:

A more comprehensive explanation of model is required, it is clearer in figure 1 description

Lines 171 – 189: A more comprehensive explanation of model is required, it is clearer in figure 1 description than in the text

Statistical analysis:

The decision about role division should have to be justified. Are there any previous studies which found role differences in evaluated parameters? If yes, please add a relevant reference here.

Discussion:

The discussion present some sentences that are too short, fragmenting linearity of text, and making it harder to read.

Conclusion:

Clear but it could be explained that more studies about different age and levels are required

Figures:

Figure 1:

Chart legend on the figure is needed to explain symbols.

I hope these comment will help to improve the overall quality of the manuscript, and I look forward to receive the revised version.

**Reviewer #2: ** The study is rigorous and innovative, with solid data and robust methods. However, I suggest including more practical applications to demonstrate the real-world impact and usability of the proposed approach.

6. PLOS authors have the option to publish the peer review history of their article (what does this mean? ). If published, this will include your full peer review and any attached files.

**Do you want your identity to be public for this peer review?** For information about this choice, including consent withdrawal, please see our Privacy Policy .

Reviewer #1: No

Reviewer #2: No

---

## [Author Response · Author response to Decision Letter 1]

28 Feb 2025

Response to comments of Reviewer #1

Title: The investigated population is composed by male youth elite soccer players. This should be indicated in title.

Agreed. We have changed the title accordingly.

Introduction: The introduction presents some sentences that are too short, fragmenting linearity of text. In addition, many phrases has no references, that should be added. In particular the following points require attention:

Agreed. We have revised the entire introduction to improve the flow of the text and have added additional references in several places.

Line 42: Reference required.

Agreed. Relevant references have been inserted.

Lines 43–63: Sentences too short fragmenting linearity of text and too many phrases with no reference.

Agreed. We have revised the text to improve the flow of reading. We have also added one or more references in four places.

Lines 67–70: Sentences too short fragmenting linearity of text and too many phrases with no reference.

Agreed. We have revised the text to improve the flow of reading. We have also added one reference.

Line 97: Reference required.

This sentence is a hypothesis. Consequently, there is no reference to this. We have changed the wording of the sentence to clarify that this is a hypothesis. In addition, we have inserted two additional sentences with references before the commented sentence. These two sentences are intended to substantiate our hypothesis and should make it easier to understand how we arrive at it.

Lines 102–107: References required.

This sentence is again a hypothesis. We have changed the wording of the sentence to clarify this. In addition, we have again inserted two additional sentences with references before the commented sentence to substantiate our hypothesis and to make it easier to understand how we arrive at it.

Participants: A table including descriptive statistics of anthropometric measures could be helpful to improve section clarity.

Agreed. We have included information on the weight and height of the participants in the text.

Measurements: It should be clarified if warm up protocols before tests have been standardized.

Agreed. The warm-up was standardized. We have added this to the text.

Event selection and model fitting: A more comprehensive explanation of model is required, it is clearer in figure 1 description.

Lines 171–189: A more comprehensive explanation of model is required; it is clearer in figure 1 description than in the text.

Agreed. We have revised this part of the article with the aim of making our methodical approach clearer and easier to understand.

Statistical analysis: The decision about role division should have to be justified. Are there any previous studies which found role differences in evaluated parameters? If yes, please add a relevant reference here.

It was not entirely clear to us what this comment meant. Our study is the first to determine the mean difference between a game-based and a test-based amax-vinit regression line. Consequently, no previous studies have reported non-uniform effects (i.e., different mean differences) for groups of players of different playing positions. Since our data showed such non-uniform effects, we decided to include playing position in the statistical model. In the text, we have included the statistical rationale for this, along with the references. In addition, we have added a possible explanation for the observed non-uniform effects. For this, we have also provided references. We hope that this sufficiently justifies the approach we have taken.

Discussion: The discussion present some sentences that are too short, fragmenting linearity of text, and making it harder to read.

Agreed. We have revised the discussion to improve the flow of the reading.

Conclusion: Clear but it could be explained that more studies about different age and levels are required.

Agreed. We have expanded the conclusion accordingly.

Figures: Figure 1: Chart legend on the figure is needed to explain symbols.

Agreed. We have added a chart legend.

Response to comments of Reviewer #2

The study is rigorous and innovative, with solid data and robust methods. However, I suggest including more practical applications to demonstrate the real-world impact and usability of the proposed approach.

Agreed. We have expanded the practical application section accordingly.

---

## [Editor Report · Decision Letter 1]

4 Mar 2025

Determining relative population-specific acceleration intensity thresholds in soccer using game locomotion data: Validation of a new method using data from male youth elite players

PONE-D-24-58536R1

Dear Dr. Andrey,

We’re pleased to inform you that your manuscript has been judged scientifically suitable for publication and will be formally accepted for publication once it meets all outstanding technical requirements.

Kind regards,

Stefano Amatori, Ph.D.

Academic Editor

PLOS ONE
---

## [Editor Report · Acceptance letter]

PONE-D-24-58536R1

PLOS ONE

Dear Dr. Andrey,

I'm pleased to inform you that your manuscript has been deemed suitable for publication in PLOS ONE. Congratulations! Your manuscript is now being handed over to our production team.

Kind regards,

on behalf of

Prof. Stefano Amatori

Academic Editor

PLOS ONE